# The Importance of Structural Factors for the Electrochemical Performance of Graphene/Carbon Nanotube/Melamine Powders towards the Catalytic Activity of Oxygen Reduction Reaction

**DOI:** 10.3390/ma14092448

**Published:** 2021-05-09

**Authors:** Piotr Kamedulski, Jerzy P. Lukaszewicz, Lukasz Witczak, Pawel Szroeder, Przemyslaw Ziolkowski

**Affiliations:** 1Faculty of Chemistry, Nicolaus Copernicus University, Gagarina 7, 87-100 Toruń, Poland; jerzy_lukaszewicz@o2.pl; 2Centre for Modern Interdisciplinary Technologies, Nicolaus Copernicus University, Wileńska 4, 87-100 Toruń, Poland; 3Institute of Physics, Kazimierz Wielki University, Powstańców Wielkopolskich 2, 85-090 Bydgoszcz, Poland; lukasz141196@gmail.com (L.W.); pawelsz@ukw.edu.pl (P.S.); 4Institute of Carbon Technologies, Gagarina 5/102, 87-100 Toruń, Poland; przemyslaw.ziolkowski@carbon4nano.com

**Keywords:** graphene, carbon nanotubes, melamine, composites, catalytic activity

## Abstract

In this paper, we show the carbonization of binary composites consisting of graphene nanoplatelets and melamine (GNP/MM), multi-walled carbon nanotubes and melamine (CNT/MM) and trinary composites containing GNP, CNT, and MM. Additionally, the manuscript presents results on the influence of structural factors for the electrochemical performance of carbon composites on their catalytic activity. This study contributes to the wide search and design of novel hybrid carbon composites for electrochemical applications. We demonstrate that intensive nitrogen atom insertion is not the governing factor since hybrid system modifications and porous structure sometimes play a more crucial role in the tailoring of electrochemical properties of the carbon hybrids seen as a noble metal-free alternative to traditional electrode materials. Additionally, HRTEM and Raman spectra study allowed for the evaluation of the quality of the obtained hybrid materials.

## 1. Introduction

Graphene and carbon nanotubes have excellent characteristics such as good mechanical [1,2], thermal [3,4], and electrical properties [5,6]. Due to their large specific surface area [7,8], graphene and nanotubes, as well as graphenenanotube networks, are promising materials in such applications as supercapacitors [9,10,11,12], lithium-ion batteries [13,14,15,16], lithium-sulfur batteries [17,18,19], zinc-air batteries [20,21,22,23], solar cells [24,25] and fuel cells [26,27,28,29].

The electrochemical properties of graphene and carbon nanotubes can be tailored by introducing heteroatoms into the carbon crystalline lattice, such as boron, sulfur, phosphorous, and nitrogen [30,31]. In particular, nitrogen, which has a smaller atomic radius and higher electronegativity than carbon, is a very good dopant that improves the electronic properties and wettability of the surface of low-dimensional carbons [32]. It also has a positive effect on the kinetics of the heterogeneous electron transfer reaction [33].

The most common method of synthesizing N-doped low-dimensional carbon is chemical vapour deposition (CVD), in which the carbon precursors are hydrocarbons (e.g., methane, ethylene) or organic solvents (e.g., benzene, toluene), and the source of nitrogen is pyridine [34] or ammonia [35]. Acetonitrile is also used as a precursor, which is a source of both carbon and nitrogen [36]. A powerful method for creating porous material is nanocasting, in which hard templates are used to create replicas with specific functionality [37]. Using this method N-doped carbon has been successfully synthesized by pyrolysis of phenanthroline in porous silica film [38,39]. Another approach is to carbonize a mixture of graphene and melamine at 800 °C. Melamine (C_3_H_6_N_6_) contains 67 wt.% of nitrogen and decomposes at 343 °C. Above the decomposition temperature, condensation reactions take place with the release of ammonia [40]. During the condensation reaction, intermediate products such as melamine, melem and melon are formed. They are more stable than melamine. The decomposition temperature of melam is ~350 °C, melem ~−450 °C, and melon ~−600 °C [41]. A further increase in temperature causes the formation of layers of thermally stable graphite carbon nitride g-C_3_N_4_ [42]. Both the release of ammonia and the formation of graphite carbon nitride as the final product of the thermalization are important from the point of view of the production of hybrid carbon structures with a large admixture of nitrogen.

Yan and coworkers [43] have shown that carbonization at 800 °C of graphene oxide and melamine with an admixture of Co, Mn and Ni oxide resulted in the growth of the N-doped carbon nanotubes (N-CNTs) on graphene oxide nanoplatelets. The surface area of the hybrid material was estimated to be 226 m^2^/g and was much higher than the surface area of the pristine graphene oxide (58 m^2^/g). 3D hybrid structures were also fabricated by pyrolysis of melamine in nickel foam, which played a double role as the platform for nanocarbon growth and catalyst [44]. This method allowed for the fabrication of the nitrogen-doped nano-carbon material with specific surface area of the hybrid material of 74 m^2^/g. In the mentioned reports the metal catalysts play an essential role in the fabrication of the nanocarbon electrocatalysts obtained from the thermal decomposition of melamine. This significantly increases the cost of producing nanocarbon-based electrode materials. 

In this paper, we subject to carbonization binary composites consisting of graphene nano-platelets and melamine (GNP/MM), multi-walled carbon nanotubes and melamine (CNT/MM), and trinary composites containing GNP, CNT and MM in order to assess the potential of the cost-effective fabrication of novel nanocarbon electrode materials without using metal catalysts. Presented results show that the simple thermalization of mixed nanocarbon/melamine powders in inert gas atomosphere resulted in significant increase in the surface area and electrochemical performance. 

As demonstrated in previous texts, electrode material properties and practical applicability are predominantly affected by their pore structure and the presence of heteroatoms. Most of the studies consider the influence of these factors separately, attributing any progress in electrode performance either to appropriately tailored pore structure or alternatively to a high level of nitrogen doping. In particular, the last factor is described as a governing one, while the role of the pore structure is somehow forgotten or even omitted. Thus, many studies are conducted on the quantitative effects of nitrogen doping, hoping to achieve possibly the highest nitrogen content. Often, it is expected that such N-doped carbons will automatically exhibit outstanding electrode properties such as the ability to effect efficient oxygen reduction reaction (ORR), which is required in some electrochemical energy storage devices such as fuel cells and air-metal batteries. The authors intend to prove the importance of the structural factor (pore structure and surface area) in the case of carbon electrode materials intensively doped with nitrogen. Additionally, the study aims at a possibly most facile synthesis of nitrogen-rich carbon hybrids containing a considerable amount of highly ordered carbon structures based on sp^2^ hybridization of carbon atoms (carbon nanotubes and graphene flakes). 

In this way, the authors intend to verify the common assumption that intensive insertion of nitrogen atoms and the presence of condensed aromatic carbon rings should be enough to obtain valuable electrode materials for ORR.

## 2. Materials and Methods

### 2.1. Materials

Commercial graphite foil with a purity of 99% was purchased from Sinograf S.A. (Torun, Poland). Mutli-walled carbon nanotubes (a diameter of 10–30 nm, a length of 5–20 µm length, nanotube content of 99 wt.%) and single-walled carbon nanotubes (average diameter of 1.6 nm, average size of 5 µm, nanotube content 75%) were supplied by the Institute of Carbon Technology Sp. z o.o. (Torun, Poland). Other reagents such as sulfuric acid (95%), melamine (Acrōs Organic, a purity of 99%), and ethyl alcohol (96%) were obtained from Chempur (Piekary Slaskie, Poland).

### 2.2. Composite Preparation

Graphene nanoplatelets (GNP) were fabricated by the electrochemical exfoliation of graphite foil. The graphite foil was cut into slices with dimensions of 3 cm × 7 cm and then dipped into a beaker containing 1.2 M aqueous solution of sulfuric acid (100 mL distilled water and 7.5 mL sulfuric acid). The graphite foil slice served as the anode, another graphite slice was used as the cathode with a DC power supply of 6 V and current up to 10 A. After 15 min reaction, the obtained dispersion of GNP in the electrolyte was vacuum filtered (micro-glass fiber filter by Ahlstrom-Munksjö Munktell, 90 mm diameter) and vacuum dried at a temperature of 120 °C for three hours.

A planetary ball mill Pulverisette 6 made by Fritsch was used to grind the components of composite materials. A 0.5 L hardened stainless steel bowl was used for grinding, the collision medium was monosized, hardened stainless steel balls with a diameter of 5 mm. The binary composite powder of GNP and MM was prepared as follows: 200 mg of GNP and 1 g of MM were mixed with 200 g of stainless steel balls and placed into the bowl. The grinding was performed under dry conditions; grinding speed and time were adjusted to 200 rpm and 2 h, respectively. To obtain the CNT/MM composite powder, 200 mg of carbon nanotubes were mixed with 1 g of MM. The trinary composition of GNP, CNTs and MM was made from 150 mg of GNP, 50 mg of MWCNTs, and 1 g of MM. For grinding the CNT/MM and GNP/CNT/MM mixtures, the same processing parameters were used as for GNP/MM.

Composite powders, referred to as GNP/MM-AsP, CNT/MM-AsP, and GNP/CNT/MM-AsP, were carbonized at 300 °C and 800 °C for two hours in the inert gas atmosphere (nitrogen) using the tube furnace Thermolyne F21100 (NIST, Gaithersbur, MD, USA).

### 2.3. Material Characterization

The obtained materials were analyzed on the equipment available at the Faculty of Chemistry of the Nicolaus Copernicus University in Torun and extensively described in our previous work [45]. Firstly, composite powders were tested by the low-temperature adsorption of nitrogen method. The relevant isotherms of all the samples were measured at −196 °C using an automatic adsorption instrument ASAP 2010 (Micromeritics, Norcross, GA, USA). Before the gas adsorption measurements, the carbon materials were outgassed under a vacuum at 200 °C for 1–2 h. The specific surface area (S_BET_) was determined based on the Brunauer–Emmett–Teller (BET) method from nitrogen adsorption data (the relative pressure range from 0.02 to 0.2). The micropore volume (V_mi_) was calculated by using only the t-plot method. Next, the total pore volume (V_t_) was determined from the amount of gas adsorbed at the relative pressure of 0.99. The volumetric elemental composition of the three main elements (carbon, nitrogen and hydrogen) of the materials was analyzed through a combustion elemental analyzer (Vario MACRO CHN, Elementar Analysensysteme GmbH, Langenselbold, Germany). The composite powders’ morphology was analyzed by scanning electron microscopy (SEM, 1430 VP, LEO Electron Microscopy Ltd., Oberkochen, Germany). Composites were also tested by high-resolution transmission electron microscopy (HRTEM, FEI Europe production, model Tecnai F20 X-Twin, Brno, Czech Republic). The carbon materials obtained before the HRTEM microscopic analysis were dispersed in ethanol 99.8% and treated with an Inter Sonic IS-1K bath for 10–15 min and deposited on holey carbon-coated copper grids. 

Raman spectra were carried out in backscattering geometry, with a Senterra Raman microscope (Bruker Optik, Billerica, MA, USA), using a ×50 objective and a 532 nm laser beam focused onto a spot of approximately 1 µm in diameter. Laser power was maintained in the range of 200 µW–20 mW. All spectra were acquired at ambient temperatures. 

### 2.4. Electrochemical Measurements

The electrochemical studies were performed according to the prepared method published in our other articles [20,46]. Electrochemical measurements were executed using an Autolab workstation (PGSTAT128N, Utrecht, The Netherlands), with a standard three-electrode system, in an electrolyte solution of 0.1 M KOH at room temperature (KOH was purchased from POCH Gliwice, Poland, KOH content min. 85%) [20,46]. Next, according to our preparation method, a glassy carbon electrode (GCE, 5 mm diameter), an Ag/AgCl electrode in 3 M KCl, and Pt wire were used as a working electrode, reference electrode, and counter electrode, respectively [20]. A commercial platinum on graphitized carbon Pt/C (20 wt.% Pt) catalyst was acquired from Sigma-Aldrich, Polish branch. 2.5 mg of the catalyst was suspended in 0.4 mL of distilled water, isopropanol, and Nafion (0.5 wt.% Nafion, Sigma-Aldrich) to form a homogeneous ink through ultrasonication for 60 min [20,46]. Then, 15.63 μL of the catalyst ink was dropped using a pipette onto the pre-polished glassy carbon electrode (GCE) and well-dried. The catalyst loading amount was approximately 0.5 mg cm^−2^ [20]. The ORR catalytic activity of the obtained samples was evaluated via cyclic voltammetry (CV) at a scan rate of 10 mV s^−1^, as well as linear sweep voltammetry (LSV) at a scan rate of 5 mV s^−1^, in an electrolyte solution of 0.1 M KOH, using an RDE system (Metrohm, Utrecht, The Netherlands) [20,46]. The LSV curves were obtained at various rotating speeds, from 800 to 2800 rpm, in an oxygen-saturated electrolyte solution. All the experiments were acquired at room temperature. The number of electrons (n) involved in the reaction can be deduced from the Koutecky–Levich (K–L) equation [20,46]:
(1)
J^−^^1^ = J_L_^−^^1^ + J_K_^−^^1^ = (Bω^1/2^) ^−^^1^ + J_K_^−^^1^
(2)
B = 0.62nFC_0_(D_0_)^2/3^ν^−^^1/6^
where J, J_L_, and J_K_ are the measured, diffusion limiting, and kinetic current density; ω is the angular velocity of the disc; n is the number of transferred electrons in the reaction; F is the Faraday constant (96,485 C mol^−1^); C_0_ is the concentration of dissolved oxygen in the 0.1 M KOH (1.2 × 10^−6^ mol L^−1^); D_0_ is the diffusion coefficient of dissolved oxygen in the 0.1 M KOH (1.9 × 10^−5^ cm^2^ s^−1^); and ν is the kinetic viscosity of the 0.1 M KOH (0.01 cm^2^ s^−1^) [20,46,47].

According to Equations (1) and (2), the number of electrons in the oxygen reduction reaction can be obtained from the slope of the K–L plot [46]. All the ORR currents presented in the figures are Faradaic currents, after correction for the capacitive current. For Tafel plots, the kinetic current was determined after mass-transport correction of RDE curves by [20,46]:

(3)
J_K_ = (J × J_L_) / (J_L_ − J)


## 3. Results

Figure 1 shows SEM images of the starting materials and as-prepared powders obtained by grinding mixtures of GNP, CNT and MM. Grinding GNP and MM resulted in a powder with a particle size of less than 2 microns (Figure 1d). The size of the graphene platelets is clearly larger than that of the melamine grains. The layered structure of graphene grains is visible. Bright areas on the GNP surface are due to the accumulation of the surface charge that is probably a consequence of the deposition of a thin layer of MM on graphene.

The grain sizes in the grained mixture of nanotubes and melamine are smaller than 1 micron (Figure 1e). As a result of mixing, a small fraction of nanotubes was incorporated into the melamine grains; the vast majority of nanotubes remained free-standing. The morphology of the ternary mixture of GNP, CNT and MM is similar to that of GNP and MM (Figure 1f). Besides GNP platelets and MM grains, both the free-standing CNTs as well as CNTs deposited on GNP are visible. The tendency to deposit on GNP is probably caused by the van der Waals interaction between the graphene planes and the nanotube cylinders.

To investigate the spatial distribution of carbon, nitrogen and oxygen, SEM-EDS mapping was made for each as-prepared composite powder. Figure 2 shows C, N, and O images at a magnification of ×300. The CNT/MM composite powder was most homogeneous. In all the samples, a surface distribution of carbon is correlated with nitrogen. The O map of the pristine GNP/MM is rough suggesting heterogeneous oxidation of graphene during electrochemical exfoliation of graphite.

The average surface mass concentration of C/N/O was found to be 25%/65%/9%, 25%/66%/8% and 11%/56%/33% in GNP/MM, CNT/MM and GNP/CNT/MM powder, respectively. It should be noted that for melamine alone, it is 28%/67%/0%. The surface mass concentration of the main abundant elements varied greatly in the GNP/CNT/MM powder. The high concentration of oxygen with a simultaneous relatively low concentration of carbon is an anomaly that is difficult to explain. 

The mass fraction of C and N obtained by elemental analysis of bulk as-prepared powders differs from the EDS analyses and was found to be 34.5%/59.1%, 35.9%/58.8% and 33.6%/60.8 in GNP/MM, CNT/MM and GNP/CNT/MM powder, respectively. As we show in Figure 3, heating at 300 °C resulted in a decrease of the nitrogen and hydrogen content. The most significant loss of nitrogen (by 26%) is observed in CNT/MM powder. On the other hand, carbonization at 800 °C led to a radical decrease in nitrogen content. In the mixture of MM and CNT, the nitrogen content was only 0.193%. It should be noted that only a slightly lower nitrogen content was found in the raw nanotubes (0.11%). It should be also mentioned that the size of the melamine grains in the CNT/MM mixture (Figure 1b) was the smallest. We observed a decrease in the hydrogen content in all the samples when the carbonization temperature increased.

Changes in the specific surface area of the powders are shown in Figure 3b. Ground mixtures of GNP, CNT and MM show a lower specific surface area than GNP (38.0 m^2^/g) and CNT (74.0 m^2^/g). The *S*_BET_ of as-prepared GNP/MM, CNT/MM and GNP/CNT/MM was found to be, respectively, 6.2 m^2^/g, 10.1 m^2^/g, and 3.5 m^2^/g. We did not observe any significant changes in the specific surface area after carbonization at 300 °C. Only the complete thermal decomposition of melamine at 800 °C caused a significant, more than ten-fold increase in the specific surface area. We obtained the best result for the CNT/MM-800 composite (104.7 m^2^/g).

The morphology of the composite powders carbonized at 300 °C and 800 °C was investigated by TEM with different magnifications. Figure 4a shows the TEM image of the GNP/MM carbonized at 300 °C, which revealed that nanoplatelets were transparent and had the structure of crumpled paper. The high-resolution TEM image of the selected area shows the wrinkled morphology of the nanoplate. On the edge of the nano platelet, a well-ordered structure with the lattice spacing of ~0.35 nm is seen, which reflects the π–π stacking interaction between graphene sheets in the platelet. Clear changes in morphology are visible if melamine is thermally decomposed at 800 °C (Figure 4b). As a result of carbonization, graphite carbon nitride (g-C_3_N_4_) was formed in the form of nanocrystals with sizes of tens of nanometers. The layered structure of the formed g-C_3_N_4_ is clearly seen in the high-resolution TEM. A side effect of melamine decomposition was the etching of the edges of the graphene platelets by ammonia (a by-product of the MM decomposition), making their texture very irregular.

When the as-prepared CNT/MM powder was heated at 300 °C, melamine was maintained as a melamine phase that did not interact with CNT. As a result, the two components of the powder did not form common structures. Figure 4c shows the CNT fraction of the powder, which consists of chaotically entangled nanotubes. The presence of melamine in the CNT fraction also cannot be detected at higher magnifications. This behavior proves that carbonization at 300 °C is not a sufficient condition for the interaction between CNT and MM to occur. Significant morphological changes occurred in CNT powder after carbonization at 800 °C. The nanotubes are much more tangled as a result of the appearance of graphite carbon nitride, which interacts with the nanotubes.

Carbonization at 300 °C of the ternary GNP/CNT/MM powder also did not lead to the formation of structures with a new morphology. Melamine grains do not tend to form common structures with nanocarbons below their melting point. On the other hand, carbon nanotubes are very eager to deposit on the graphene surface by van der Waals forces. Completely new features became apparent after carbonization at 800 °C. The edges of the graphene are jagged due to etching by ammonia. Both carbon nanotubes and graphene form common structures with the g-C_3_N_4_.

Figure 5 shows Raman spectra of the as-prepared and carbonized powders. Apart from the sp^2^-carbon D, G and 2D bands, the spectra of the as-prepared powders also show bands derived from melamine. The bands at 376 cm^−1^, 669 cm^−1^ and 980 cm^−1^ are assigned to, respectively, δ(CN), ring breathing, and δ(CNC) + δ(NCN) [48]. MM bands disappeared after carbonization at 300 °C. Instead, a strong fluorescent background appeared that could obscure the melamine bands. The strong fluorescence background disappeared after carbonization at 800 °C.

The position and relative intensity of the sp^2^-carbon D, G and 2D band, which are summarized in Table 1, provide information about the doping and defects. Both in the case of graphene and nanotubes, a blue shift in the G and 2D-mode features was observed after grinding the carbon components with melamine. Blue shift of the main Raman bands disappeared after carbonization of the powders. Note that grinding the ternary mixture of GNP, CNT and MM did not cause a significant G and 2D line shift.

If the shift of the Raman bands is due to lattice stress only, the slope of the correlation plot $\Delta \omega_{G}^{\mathrm{strain}}\;\mathrm{vs}.\;\Delta \omega_{2D}^{\mathrm{strain}}$ is 2.21 [49]. Pure doping causes a much smaller correlation slope, which is ~0.5 and ~0.3 for low concentrations (<$5\times 10^{12}{\;\mathrm{cm}}^{-2}$) of electrons and holes, respectively [50]. With this in mind, we conclude that the grinding of GNP and MM caused electron transfer from MM to GNP ($\Delta \omega_{2D}/\Delta \omega_{G}\approx 0.25$). A more significant shift of the 2D-mode in relation to the G mode was observed after sample carbonization at 300 °C, which proves the compressive strain formation in the graphene network. Annealing the defects during carbonization at 800 °C makes the occurring stresses weaker (lower the 2D-mode shift).

CNTs and CNT/MM powders showed the clear G and 2D-mode shift correlation with a 1.45 slope. This proves that mixing nanotubes with melamine and carbonization caused, first of all, carbon sp^2^ lattice strain. In the ternary powders, the charge transfer between the components determined the changes in the G and 2D-mode positions (low shifts of the 2D-mode in comparison with the shifts of the G-mode). In this case, mutual doping between CNT and GNP occurs [51].

We observed changes in the relative intensities of the G and 2D-mode features induced during each stage of composite fabrication. The relative intensity of the 2D-mode of GNP decreased after the grinding process with melamine. Subsequent carbonization resulted in a significant increase in intensity. On the other hand, we did not observe such a strong influence of the particular treatment steps on the 2D-mode intensity in the binary mixture of CNT and MM. The three-component composition was the most sensitive to mixing and carbonization. The relative intensity of the 2D line dropped by more than half after carbonation at 300 °C. A slightly smaller decrease occurred after carbonization at the temperature of 800 °C.

To assess the point-like defect concentration, $n_{d}$, we used semi-empiric formula [52]
(4)$$n_{d}=\frac{1.8\times 10^{22}}{\lambda_{L}^{4}}\left(\frac{I_{D}}{I_{G}}\right),$$
where $\lambda_{L}$ is the excitation laser wavelength (in nm). Extremely high defect density ($n_{D}>1.5\times 10^{11}\;{\mathrm{cm}}^{-2}$) shows GNP and as-prepared GNP/MM powder. High defect concentration is a consequence of the electrochemical exfoliation process of graphite foil in the aqueous solution of sulfuric acid. A side effect of the graphite exfoliation process is partial oxidation of the graphene flakes. However, the defects were annealed during the carbonization of the GNP/MM powder. Heating at a temperature of 300 °C leads to a four-fold decrease in the defect concentration. Carbonization at 800 °C is more efficient; we observed a five-fold decrease in defects. We did not observe such a trend in CNT/MM and GNP/CNT/MM samples. A slight increase in the defect concentration is related to the formation of carbon nitride, which is a product of the thermal decomposition of melamine.

In turn, the catalytic activity of representative obtained carbon composites was investigated using the techniques of cyclic voltammetry (CV) and linear sweep voltammetry (LSV). All measurements were made in 0.1 M KOH saturated with oxygen and nitrogen. Figure 6a shows the CV curves from which a clear cathode peak (Ep) is visible in the obtained graphene materials. The cathode peak is clear for all obtained samples. Table 2 shows the oxygen reduction reaction parameters necessary to characterize the catalytic activity of the electrocatalyst. The value of the cathode peak for commercial platinum-based carbon (Pt/C, 20% wt.) is 0.76 V vs. RHE, while for the representative samples, clear peaks are observed at 0.77 V for CNT/MM-300 and 0.78 V for CNT/MM-800 sample. Further linear sweep voltammetry (LSV) measurements were performed with a rotating disc electrode (RDE) in the rpm range from 800 to 2800 rpm in an O_2_ saturated electrolyte of 0.1 M KOH. Table 2 shows the starting potential values for each sample at 1600 rpm resulting from the linear RDE sweep (Figure 6b). The onset potential (E_onset_ in Table 2) for the ORR is an important criterion in evaluating the activity of an electrocatalyst. An onset potential lower than that of a Pt/C catalyst was registered. The CNT/MM-300 sample shows the onset potential of 0.76 V. Nevertheless, the CNT/MM-800 sample show the most positive onset potential of 0.77 V. 

The number of transferred electrons (Table 2 and Figure 6c) for most of the obtained carbon materials is above 3. The highest value of the number of electrons transferred for the CNT/MM-800 sample is 2.80, which indicatesa dissimilar 4-electron oxygen reduction reaction to the case of a commercial catalyst Pt/C. However, the tendency to the increase in onset potential, E_s_, halfwave potential, E_1/2_, diffusion limiting current and the averaged electron transfer number for ORR with increasing carbonization temperature is very clear. This finding is linked to the position of the G and 2D Raman mode, which is less red shifted with respect to the raw CNTs in CNT/MM mixture carbonized at 800 °C than in 300 °C. Red shift of these Raman features proves the presence of tensile stress in carbon nanotubes [53]. The decrease in stresses as a result of carbonization at 800 °C is correlated with an almost ten-fold increase in the specific surface area of the composite resulting from thermal decomposition of MM. On the other hand, a slight increase in the relative intensity of the D-band in the CNT/MM mixture carbonized at 800 °C is caused by an increase in the defect concentration by about 16% in relation to the mixture annealed at 300 °C. We attribute the additional defects to the formation of g-C_3_N_4_ nano-islands visible in the HRTEM images (Figure 4d). The separated g-C_3_N_4_ nano-islands contribute to a slight increase of the nitrogen impurities from 0.11 wt.% in pristine CNTs to 0.19 wt.% in MM/CNT-800. The surface area and porosity of the obtained composite materials allow for unforced access of the KOH electrolyte into the porous structure, increasing the catalytic activity. Therefore, these materials can be used as metal-free catalysts in devices based on the oxygen reduction reaction, such as metal-air batteries or fuel cells. The porosity of the graphene structure has a significant impact on the catalytic activity and the appropriate pore size means that materials with a high surface area can also be used in energy storage devices.

Based on the results of the oxygen reduction reaction in the representative samples, the key parameter, that is, the number of electrons (n) participating in the reaction ORR, is on the level from 2.77 to 2.80. In the case of the starting carbons (Figure 6c), the average electron transfer number of electrons is 2.00 for graphite, 2.17 for pristine CNT and 2.06 [54] for g-C_3_N_4_. Thus, an increase in the number of n for the obtained composites is visible. Moreover, noticeable is the lack of the effect of temperature on the value of n.

## 4. Conclusions

In summary, we have demonstrated that carbonization of nanocarbon/MM mixtures is a promising route for obtaining electrocatalytic materials with high surface area and high electrocatalytic activity without the participation of metal catalysts. We have shown that carbonization of GNP/MM, CNT/MM, GNP/CNT/ resulted in the increase in surface area.

Additionally, we have demonstrated the influence of some key parameters such as porous structure, the introduction of heteroatoms, carbonization temperature and modification of structure on the properties of carbon hybrids seen as a candidate for an electrode material catalyzing oxygen reduction. Not all of these key parameters have a positive effect on the electrochemical performance of the carbon hybrids in ORR. Despite intensive nitrogen doping and the presence of condensed aromatic rings, the hybrids showed only moderate activity in ORR. The tested carbon hybrids suffered from the lack of well-developed surface area and pore structure. The obtained results highlight the importance of structural properties of potential electrode materials, the lack of which leads to the inactivation of N-originated catalytic centers definitely existing on the carbon surface. This negative effect was observed despite the high content of condensed carbon aromatic rings, which theoretically should have provided a high concentration of movable π-electrons. The problem of the importance of structural parameters in ORR on carbon electrodes will be investigated in further studies.

## Figures and Tables

**Figure 1 materials-14-02448-f001:**
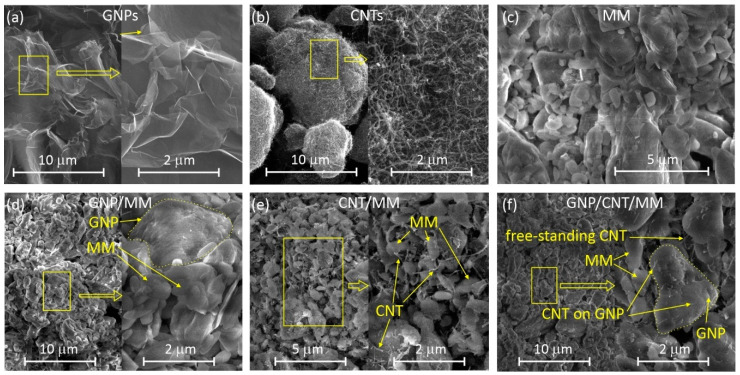
SEM images of the starting materials and as-prepared powders: (**a**) GNPs, (**b**) CNTs, (**c**) MM, (**d**) GNP/MM, (**e**) CNT/MM, (**f**) GNP/CNT/MM.

**Figure 2 materials-14-02448-f002:**
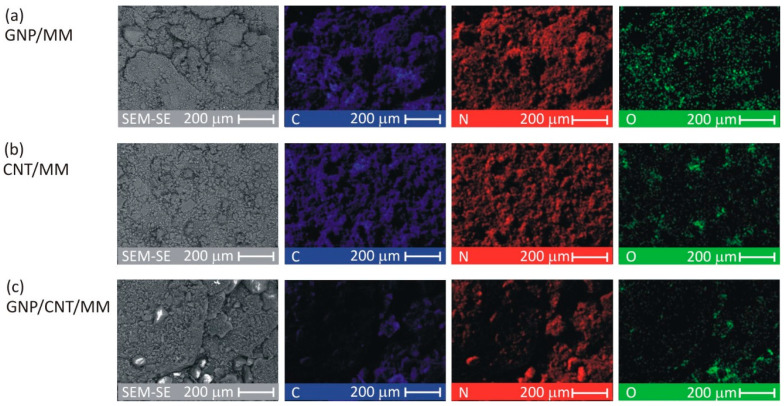
SEM-EDS images of the as-prepared composite powders.

**Figure 3 materials-14-02448-f003:**
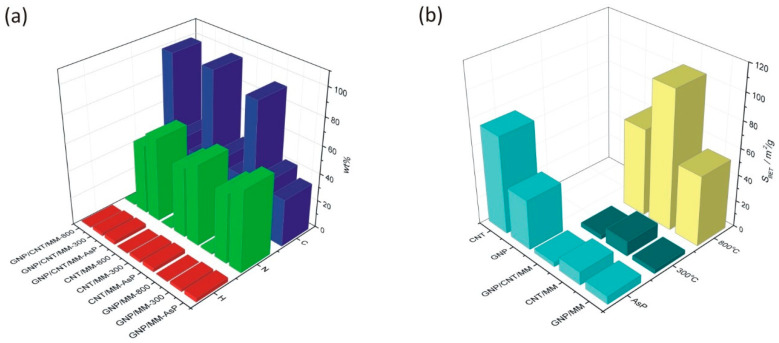
(**a**) Mass fraction of H, C and N in as-prepared powders and composites subjected to carbonization at 300 °C and 800 °C. (**b**) Changes in the specific surface resulted from the carbonization process.

**Figure 4 materials-14-02448-f004:**
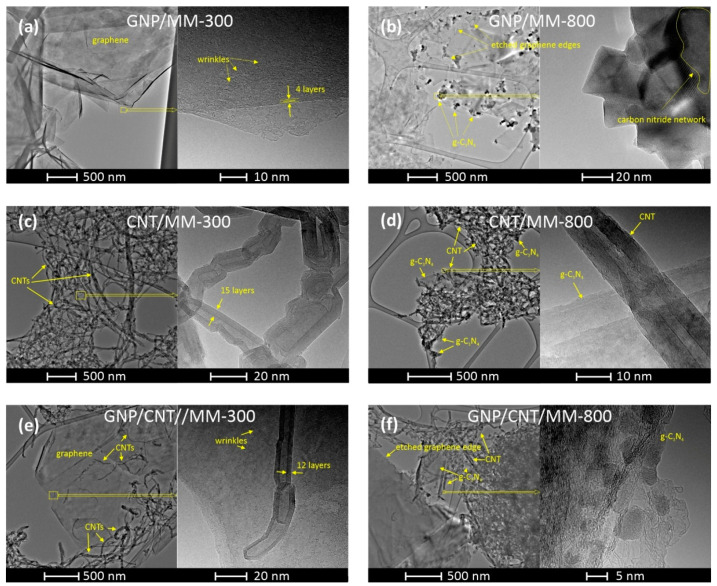
TEM images of GNP/MM (**a**,**b**), CNT/MM (**c**,**d**), and GNP/CNT/MM (**e**,**f**) composites carbonized at 300 °C and 800 °C.

**Figure 5 materials-14-02448-f005:**
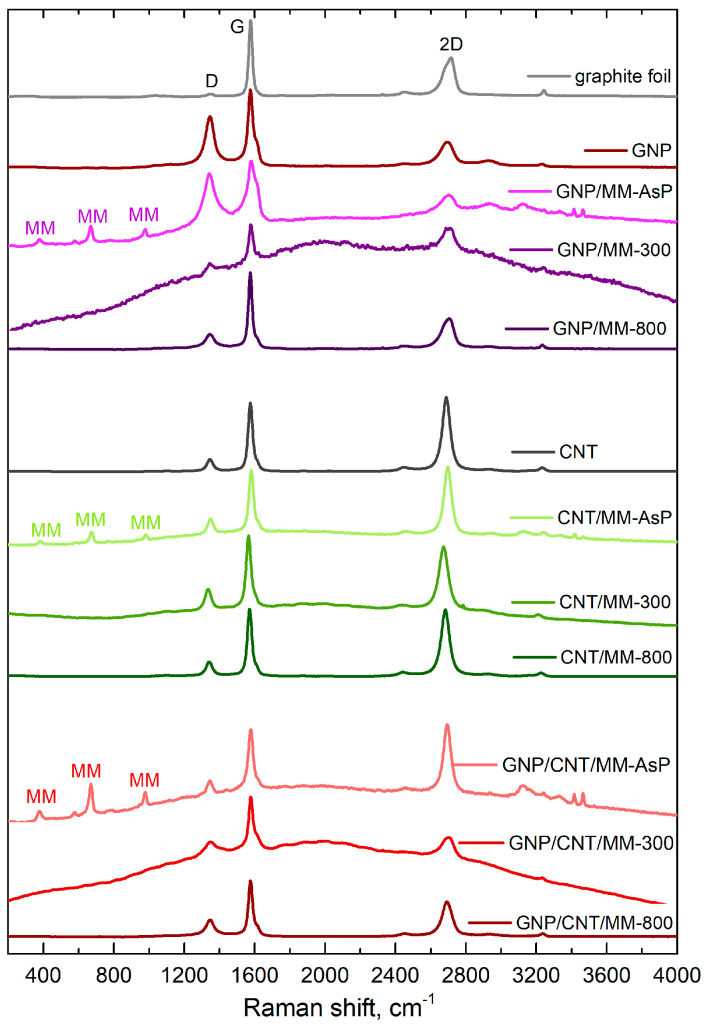
Raman spectra of composite powders. The main sp^2^-carbon Raman bands are assigned as D, G and 2D. The Raman bands derived from melamine are signed as MM.

**Figure 6 materials-14-02448-f006:**
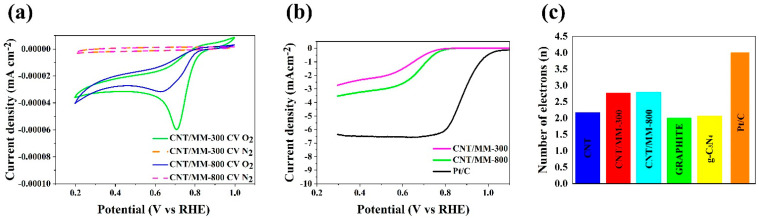
(**a**) CV curves of obtained electrocatalysts in an N_2_ and O_2_-saturated 0.1 M KOH solution; (**b**) LSV curves of CNT/MM-300, CNT/MM-800 and Pt/C catalysts measured at a scan rate of 5 mV s^−1^ and a rotation rate of 1600 rpm in O_2_-saturated 0.1 M KOH solution; (**c**) number of transfer electrons in the oxygen reduction reaction.

**Table 1 materials-14-02448-t001:** Parameters of the Raman scattering lines.

Sample	D (cm^−1^)	G (cm^−1^)	2D (cm^−1^)	*I*_D_/*I*_G_	*n*_d_ (×10^10^ cm^−2^)	*I*_2D_/*I*_G_
graphite foil	1348.5	1576.28	2719.52	0.027	0.62	0.331
GNP	1344.6	1576.43	2691.29	0.680	15.27	0.326
GNP/MM-AsP	1343.1	1586.06	2693.84	0.797	17.91	0.264
GNP/MM-300	1349.6	1578.15	2700.87	0.233	5.24	0.658
GNP/MM-800	1345.4	1574.71	2697.40	0.181	4.06	0.411
CNT	1345.4	1575.35	2688.09	0.172	3.87	1.08
CNT/MM-AsP	1348.4	1580.33	2696.45	0.251	5.63	1.106
CNT/MM-300	1334.8	1566.35	2675.88	0.178	4.01	0.907
CNT/MM-800	1341.1	1571.04	2682.73	0.208	4.67	0.985
GNP/CNT/M-AsP	1343.5	1576.90	2693.32	0.225	5.06	1.170
GNP/CNT/M-300	1351.2	1578.34	2693.28	0.255	5.72	0.434
GNP/CNT/M-800	1345.8	1576.23	2692.56	0.284	6.38	0.640

**Table 2 materials-14-02448-t002:** ORR performance parameters of representative samples and commercial Pt/C catalysts tested in alkaline media.

Catalysts	E_p_ (V vs. RHE)	E_onset_ (V vs. RHE)	E_1/2_ (V vs. RHE)	Diffusion-Limiting Current (mA cm^−2^)	n (0.5 V)
Pt/C (20% wt.)	0.76	1.030	0.88	6.37	4.00
CNT/MM-300	0.77	0.759	0.72	3.08	2.77
CNT/MM-800	0.78	0.772	0.74	3.26	2.80

## Data Availability

The data presented in this study are available on request from the corresponding author.

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
