# Peer review of "The Importance of Structural Factors for the Electrochemical Performance of Graphene/Carbon Nanotube/Melamine Powders towards the Catalytic Activity of Oxygen Reduction Reaction"

_materials, 2021, doi:10.3390/ma14092448_

Round 1
Reviewer 1 Report
The overall English needs to be improved, examples including line 41 "hardly N-doped low-dimensional carbon" should be avoided.
Fig.1. The reference SEM image of GNP, CNT and MM should be provided for comparison.
Please explain the low concentration of C element for GNP/CNT/MM in Fig.2.
In general, graphene and CNT have high specific surface areas, the composite materials that authors made have significantly lower surface areas, which will not be beneficial for catalysis. In summary, I did not see the novelty from the current work, the improvement in catalytic performance could be caused by the difference in surface area.
Reviewer 2 Report
The Importance of structural factors for the electrochemical performance of graphene/carbon nanotube/melamine powders on its catalytic activity towards ORR
The authors report the electrochemical activity toward ORR of various composites made of the combination of graphene nano pellets, carbon nanotubes and melamine. After different combinations, mechanical milling, and thermal treatments, they evaluated the electrochemical performance of the oxygen reduction reaction. The authors pretend to demonstrate the influence of structural factor on the electrocatalytic activity. They concluded that the insertion of nitrogen did not play a critical influence in the activity, rather, the modification of the pore structure can be more crucial.
In general, the structure of this manuscript is clear, however some typos were detected along the manuscript. The experimental section and methodology are well described and clear, however the discussion of results and the connection among the physicochemical characterization with the electrochemical results was very vague and it was not well achieved.
The introduction was focused on the different methods of synthesis and the insertion of nitrogen within carbon. The authors failed to present prior studies related to carbonaceous electrocatalysts applied to the ORR in alkaline media. The inclusion of nitrogen within the carbon structure have been probe very important for increasing the electrochemical activity. This fact was not mentioned at all and I consider this as a relevant information, in regards that the authors suggested that the inclusion of nitrogen was not important in this work. A more thorough bibliographic analysis in this direction is recommended.
Recommendation 1: I would suggest changing the title to “The Importance of structural factors for the electrochemical performance of graphene/carbon nanotube/melamine powders towards the catalytic activity of oxygen reduction reaction.”
Several typos are observed in manuscript and tables. Authors should double-check the overall manuscript.
Line 36-37: phosphor by phosphorous
Line 67: graphene fakes by graphene flakes
Line 110: The total pole volume by The total pore volume
Line 114: not sure what xxx means
Line 133: what content of Nafion ionomer was used?
Line 136: According to the numbers reported, my estimation was about 0.5 mg/cm2.
Line 194: I don’t understand what the authors want to say. “ The more that is not confirmed by elemental analysis.”
In table 2: change 0.7,59 by 0.759
Line 205-206: I would suggest changing the following sentence “We observed a decrease in the hydrogen content in all the samples. The lower the hydrogen content, the higher the carbonization temperature.” By “We observed a decrease in the hydrogen content in all the samples when the carbonization temperature increased.”
- In terms of the elemental content distribution, could the authors include the standard deviation of the statistical analysis?
- During the characterization by HRTEM, the figure 4f presents large number of segregated particles labeled as C3N4, however the average content of nitrogen within the sample was reported very low.
Is this image representative of the general condition of the sample?
How would this segregation affect the electrochemical activity?
- The discussion of the characterization by RAMAN is completely disconnected from the electrochemical analysis and vice versa. I could not find any relationship between the content of nitrogen or the presence of defects with the electrochemical results.
- The authors claim for a correlation of structural factors and the activity toward ORR, however did not present the results of GNP/MM-300, GNP/MM-800, GNP/CNT/MM-300 and GNP/CNT/MM-800, which were extensively discussed during the characterization?
- I would suggest the inclusion of the ORR activity of GNP and CNT as reference.
- When the authors in line 325-326 say: “The obtained materials, despite the medium nitrogen content in the structure, show increased catalytic activity”
What materials are they referring to?
What do they mean with medium nitrogen content, assuming that the nitrogen content was reported marginal?
What parameter are the authors considering as representative of the activity?
- Can the authors clarify what factors influence the electrochemical performance and in what way?
- Can the authors include a graphic correlating any of the parameters mentioned in the manuscript, for example: content of nitrogen, surface area, pore size, pore distribution etc., with the electrochemical activity?
- Can the authors include some comparison of the activity toward ORR with other catalysts with similar characteristics?

Reviewer 3 Report
Major revisions are necessary. Please find attached the relative comments

Round 2
Reviewer 2 Report
Thank you for the changes, the article looks much better, the different subjects are now connected and better presented.
Reviewer 3 Report
Please, find attached my comments